# Controlled Transport of Magnetic Particles and Cells Using C-Shaped Magnetic Thin Films in Microfluidic Chips

**DOI:** 10.3390/mi13122177

**Published:** 2022-12-08

**Authors:** Roozbeh Abedini-Nassab, Ali Emamgholizadeh

**Affiliations:** Faculty of Mechanical Engineering, Tarbiat Modares University, Tehran P.O. Box 14115-111, Iran

**Keywords:** magnetic thin film, magnetic particle transport, microfluidic, single-cell

## Abstract

Single-cell analysis is an emerging discipline that has shown a transformative impact in cell biology in the last decade. Progress in this field requires systems capable of accurately moving the cells and particles in a controlled manner. Here, we present a microfluidic platform equipped with C-shaped magnetic thin films to precisely transport magnetic particles in a tri-axial rotating magnetic field. This innovative system, compared to the other rivals, offers numerous advantages. The magnetic particles repel each other to prevent undesired cluster formation. Many particles move synced with the external rotating magnetic field, which results in highly parallel controlled particle transport. We show that the particle transport in this system is analogous to electron transport and Ohm’s law in electrical circuits. The proposed magnetic transport pattern is carefully studied using both simulations and experiments for various parameters, including the magnetic field characteristics, particle size, and gap size in the design. We demonstrate the appropriate transport of both magnetic beads and magnetized living cells. We also show a pilot mRNA-capturing experiment with barcode-carrying magnetic beads. The introduced chip offers fundamental potential applications in the fields of single-cell biology and bioengineering.

## 1. Introduction

Studying cells, their interactions with other cells and other biological particles, and their drug responses is a key task in the field of biology [1,2,3,4,5]. For example, studying cells with an unknown disease (e.g., cancer) or cells infected with a new virus (e.g., coronavirus) is of particular importance [6,7]. Moreover, investigating the effect of a new drug (such as a new vaccine) on viruses is one of the important steps toward discovering novel treatment methods [8].

To perform the necessary tests, traditionally, a considerable volume of sample is taken, and the average response of the cells is measured. In many diseases, some rare cells determine the patient’s fate; however, finding these cells in the traditional methods is impossible. These cells are hard to find in highly heterogeneous cell populations (e.g., in cancer tumors) [9,10].

The single-cell study is considered a modern approach to dealing with this problem. In this method, even by taking much fewer samples from the body (e.g., liquid or solid biopsies) and conducting studies on a limited number of particles and cells, compared to that of the traditional methods, important rare cells can be detected and studied. Along with this concept, the lab-on-a-chip (LOC) technique has been introduced and has resulted in a revolution in medical diagnosis and treatment [11,12]. Thus, screening and diagnosis of diseases such as cancer can potentially be implemented using single-cell analysis methods based on LOC systems. Many good results are already achieved. However, towards this goal, an important step is to obtain a reliable technique to transport the particles of interest (e.g., cells or barcode-carrying beads) to the desired locations on the chip where various tests can be conducted. There are multiple methods for assembling particle arrays on chips, including hydraulic traps [13,14], acoustic traps [15,16], arrays of wells [17,18], and magnetic-based techniques [19]. The hydraulic traps may cause undesired shear stress on trapped cells. Moreover, due to the nature of the methods based on acoustic forces, they mostly work at the bulk level and not at the single-cell resolution. In methods based on arrays of wells, precise particle control is not offered. Magnetic particle manipulation is a promising method; however, in the proposed approaches based on embedded micro-coils [20,21], the wiring system is complicated. Current-carrying wires may also increase the chip temperature. Some other magnetic manipulation methods are based on external magnetic fields [22,23], which cannot provide precise control over the particles.

Precise transport of particles with the ability to predict the dynamic trajectory of them at single-particle and single-cell levels is important because it allows for designing crucial applications in biology and medicine. For example, it provides the opportunity to form arrays or cells for high-throughput single-cell studies. It also allows the forming of cell pairs and studying cell-cell interaction at high resolution. It also allows for transporting cells to various environments and studying the environmental effects of the cells. Drug screening, studying single-cell infection, and precise tissue engineering, are other applications in which precise particle and cell transport is a fundamental requirement. To achieve the mentioned goal, we have previously introduced magnetophoretic circuits [24,25,26,27,28], which are designed to operate similarly to electrical circuits with the same control and automation. However, they are adapted such that they transport tiny particles and cells as opposed to electrons. However, in the early generations of these circuits, which operate in two-dimensional (2D) in-plane magnetic fields, the particles tend to attract each other. The attractive force is due to the fact that the opposite poles of the adjacent particles are aligned toward each other (see Figure 1d). This manner is different from the behavior of electrons in electrical circuits that ripple each other. This phenomenon may ultimately result in particle cluster formation and potentially alter the device’s operation.

To overcome this challenge, here we introduce microfluidic chips equipped with C-shaped magnetic thin films operating in a tri-axial magnetic field (see Figure 1a). In this system, a vertical magnetic field is superimposed on the in-plane rotating field to bias the magnetic particles such that the dipolar interactions between them become repulsive (compare Figure 1e with Figure 1d, and see Figure 1b for the experimental setup). The opposite poles are aligned such that they do not face each other anymore. Hence, the chance of forming particle clusters is lowered. The C-shaped design is simple and compact, which allows for designing high-density circuits in which numerous particles can be transported simultaneously. It will be explained in the Results Section that the original disk-shaped magnetic patterns cannot transport the particles in the proposed tri-axial magnetic field. The proposed circuits in this work are unique among rivals because they offer a simple and compact design for precise, highly controlled single-particle manipulation, with repulsive force between the particles to prevent particle cluster formation.

We use finite element methods (FEM) to simulate the energy distribution on the introduced chip. We use the simulation results to predict the particle trajectories on the chip. We study various parameters, including the gap between the magnetic thin films and the particle size. We also investigate the effect of adding an I bar in between the C films. Based on the obtained insights, we design the magnetic track and fabricate them. We show that the experimental results agree with our predictions based on simulations.

In the next section, we explain the theory and simulation methods. We also mention how we fabricate the chips. Next, we demonstrate the simulation results and experimental achievements. After discussing the results, we conclude the work.

## 2. Materials and Methods

In this section, the theory and methods used for simulations are explained first. Then the experimental methods are presented.

### 2.1. Theory and Simulations

A magnetic thin film in an external magnetic field is magnetized and influences the surrounding magnetic energy. The magnetic energy is calculated from Equation (1) [29].
(1)$$U=\frac{1}{2}\mu_{0}V_{p}\left(\chi_{p}-\chi_{f}\right)H^{2},$$
where *µ*_0_, *V_p_*, *χ_p_*, *χ_f_*, and *H* are the vacuum magnetic permeability, particle volume, the magnetic susceptibility of the particle, the magnetic susceptibility of the liquid carrying the particle, and the magnetic field intensity, respectively. The magnetic flux density and magnetic field intensity are related with Equation (2) [29].
(2)$$\vec{B}=\mu \vec{H},$$
where *B* and *µ* are the magnetic flux density and magnetic permeability, respectively. From Equation (1), the magnetic potential energy distribution is obtained, which shows areas with minimum energy. The magnetic particles tend to move to these spots (i.e., the energy wells). The magnetic driving force can be calculated from Equation (3).
(3)$$\vec{F}=-\nabla U,$$

We use COMSOL Multiphysics 5.6 software to run the FEM simulations. We used the AC/DC module and the Magnetic Field, No Currents interface. The governing equations in this interface are based on Maxwell’s Equations, and the magnetic ones are presented in Equations (2), (4), and (5).
(4)$$\nabla .\vec{B}=0,$$
(5)$$\nabla \times \vec{H}=J+\frac{\partial D}{\partial t},$$
where *J* and *D* are electrical current and electric displacement, respectively.

The boundary conditions are the external magnetic fields that are set on the boundaries (i.e., at the faces of a container cube, with the size of 1 mm. This cube surrounds the magnetic pattern. The initial values of magnetic fields are set to zero. Iterative stationary solver is used. In these simulations, we only included the magnetic thin film, and not the silicon wafer, which is not considered a magnetic material.

After creating the geometries in 3D model, we created an appropriate mesh for each design to achieve convergence. For example, for the I bar, we created a mesh with 2,190,163 elements, 44,336 triangles, 1088 edge elements, and 24 vertex elements, and for the disk-shape magnet, we created a mesh with 466,813 elements, 17,034 triangles, 568 edge elements, and 16 vortex elements. To validate the simulation results we compared the obtained particle trajectories with the experimental achievements.

### 2.2. Experimental Methods

The schematic in Figure 1c shows the fabrication steps. First, cleaned silicon wafers (University Wafer, Boston, MA, USA) were spin-coated with negative photoresist (NFR16-D2 JSR Micro Inc., Sunnyvale, CA, USA) for 5 s at 500 RPM, followed by 30 s at 3000 RPM (Headway spinner; Garland, TX, USA). Then, after baking at 90 °C for 60 s on a hotplate, the chips were exposed to ultraviolet light for 12 s at an illumination power of 13.5 mW at a wavelength of 365 nm (Karl Suss MA6/BA6; SÜSS MicroTec, Garching, Germany). Next, the chips were baked at 90 °C for 60 s on the hotplate. Microposit MF-319 (Shipley, Marlborough, MA, USA) was used for 60 s for the development step, after which the chips were rinsed with deionized water, and dried with nitrogen gas. After plasma ashing (O_2_) for 60 s at 100 mW, a 5 nm/100 nm thick stack of Ti/Ni_80_Fe_20_ film was deposited on top of the chips using an electron-beam evaporating system (Kurt Lesker PVD 75; Kurt J. Lesker Company Ltd., Dresden, Germany), at an operating pressure of 1 × 10^−5^. To remove the excess metals, the chips were kept in 1165 resist remover (NMP) at 65 °C, for 5 min. Then, they were rinsed with acetone and isopropanol and dried with nitrogen gas. The chip surface was then coated with a 200 nm thick layer of SiO_2_, using the Plasma Enhanced Chemical Vapor Deposition (PECVD) method (Advanced Vacuum Vision 310) at 250 °C, at the rate of 35 nm/s.

In our experiments, we used a microscopy stage that is equipped with a camera to take pictures of the particles on the chip (see Figure 1b). The chip was placed under the microscope and in between five magnetic coils, four of which were located in-plane with the chip, and the fifth one sit underneath the chip. These coils provided the required rotating magnetic field and the resulting force to move the magnetic particles. The experiments were performed with Spherotech CM-50-10, FCM-8056-2, and CM-150-10 magnetic beads (Spherotech Inc., Lake Forest, IL, USA), with mean diameters of 5.7, 8.4, and 15.6 µm, respectively, and with CD4+ T cells magnetically labeled with anti-CD4 antibody-conjugated magnetic nanoparticles (StemCell Technologies, Vancouver, BC, Canada). The cell labeling was performed based on the protocol suggested by the manufacturer. Briefly, the selection cocktail was added to the single cell suspension (100 μL/mL cells), mixed, and incubated at room temperature for 15 min. Then, the magnetic nanoparticles were added to the solution (50 μL/mL cells), mixed, and incubated at room temperature for 10 min. Then, the labeled cells were collected using a magnet.

In mRNA capture experiments, deoxythymidine oligonucleotides (oligo (dT))-coupled beads were used. This protocol is based on pairing the poly-A tail of mRNA and the oligo dT sequences available on the beads. Using a magnet, the magnetic beads were collected from 100 μL (0.5 mg) of the bead suspension. Then the beads were resuspended in 500 μL Binding Buffer (ThermoFisher Scientific, Waltham, MA, USA). Then the beads were introduced to the sample solution on the magnetophoretic chips. After running the experiments, the beads were washed with Washing Buffer (ThermoFisher Scientific) to remove non-specifically bounded contaminants. Finally, the mRNA level was evaluated with polymerase chain reaction (PCR).

## 3. Results and Discussions

We assume that permalloy (i.e., Ni_80_Fe_20_) is a soft magnetic material such that its magnetization is synched with the external rotating magnetic field. A magnetic bar in an in-plane magnetic field along its long axis behaves as an induced magnet with two poles on their ends [30]. The same behavior is seen when a magnetic disk is exposed to an in-plane magnetic field. Based on simulation results, two energy wells on the magnetic poles are formed (blue regions in Figure 2a,b). Magnetic particles and cells tend to move to these areas with minimum energies. In magnetophoretic circuits operating in an in-plane magnetic field, a periodic magnetic pattern composed of magnetic disks is used. In this system, the energy wells of the two adjacent magnetic disks overlap, and the magnetic particle following one of these energy wells and circulating a magnetic disk faces the other overlapping energy well on the adjacent magnetic disk. Hence, the particle moves from one magnetic disk to the next one. Thus, in a rotating magnetic field, the particles are transported along the magnetic tracks.

Now, a superimposed vertical field turns one of the energy wells on these magnetic patterns into an energy barrier (see Figure 2c,d). Hence, any magnetic bar or disk has a single attractive pole and a single repulsive pole. Thus, in a periodic magnetic pattern, only a single energy well circulates each magnetic disk and cannot overlap with the energy well created by the adjacent magnetic disk. As a result, particle transport fails.

To overcome this challenge, after the particle circulates half the perimeter of the first magnetic disk such that it approaches the next disk, we need to hold it until the single energy well rotating the next magnet arrives (i.e., the particle moves the magnetic track during the first half cycle, and it is held for the other half cycle). This goal is achieved by turning disks into C-shaped magnetic patterns. We also add I bars between the two adjacent C-shaped magnets to hold the particle on its tip (see Figure 3).

To better study the dynamics of the particle trajectories in this transport system, we use FEM analysis to model the magnetic energy distribution at the center of the magnetic particle of interest. Figure 3 demonstrates the sequence of magnetic energy distributions when the external magnetic field rotates 360°. The blue regions in this figure stand for the areas with low energies, attracting the magnetic particles. Starting from Figure 3a, the blue regions and their following magnetic particles are located on the left side of the C-shaped magnets. While the magnetic field rotates clockwise, in Figure 3a–e, the energy well circulates the C-shaped magnets, arriving on their right side. At this point, as the external field rotates, a deep energy well is formed on the I bar tip (see Figure 3g). By further rotating the external field, a deep energy well is formed on the left side of the next C-shaped magnet on the right, and the particle moves from the I bar to that C-shaped magnet (see Figure 3h). As a result, the proposed magnetic track transports the magnetic particles in a rotating magnetic field. Each particle moves one period (i.e., from one C-shaped magnet to exactly the same spot on the next one) in each magnetic field cycle (i.e., 360°).

To better study the particle transport from the C-shaped magnet to the I bar, we plot the magnetic energy along the line connecting these two points (the BC line in Figure 4a) and compare it with that along the line connecting the two feet in the C-shaped magnet (the BA line in Figure 4a). The energy heatmap plots and the corresponding energy plots along the mentioned lines for three important field angles at which the particle switching between the magnets happens are shown in Figure 4a–c and Figure 4d–f, respectively. In Figure 4d, an energy well at point B with energy barriers on both sides holding the particle at that point is seen. Then, in Figure 4e, the energy level at point C is lowered; however, the energy barriers on both sides of point B still exist. Hence, at this field angle, the particle still remains at point B. By further rotating the external field, the energy barrier between points B and C disappears, and a negative slope toward point C is formed (see Figure 4f). The energy barrier between points B and A is still obvious, which prevents the particle from unwanted backward movement. Thus, the magnetic particle moves forward along the line BC towards point C on the I bar tip. To systematically study the particle transport, we define λ = x/AB, which is used in the x-axis of the plots in Figure 4 and the rest.

Similarly, to investigate the particle movement from the I bar to the next C-shaped magnet on the right, we plot the magnetic energy along the lines connecting the I bar tip to the two C-shaped magnets on both sides (points A, B, and C in Figure 5a). In Figure 5d, the deep energy well at point B implies that the magnetic particle is located on the I bar tip. Then, by rotating the field, in Figure 5e, the energy barrier along the BC line becomes smaller, and in Figure 5f, it disappears. A negative slope towards point C (see Figure 5f) indicates that the magnetic particle moves forward along the BC line towards point C on the I bar tip. Please note that the positive slope along the BA line prevents the particle from unwanted backward movement.

The cone angle, frequency, and magnitude of the external magnetic field are three important parameters to be adjusted in the experiments. Thus, we first study their effects to find their appropriate ranges. The experimentally achieved efficiency of the proposed design in transporting magnetic particles (i.e., the ratio of the proper transportation in each cycle to the total particle transportations) for multiple magnetic field cone angles in the range of α = 26–63° at various frequencies are plotted in Figure 6a–c. Based on these results, α = 30–60° is considered the proper cone angle range for the external magnetic field.

Moreover, the particles are transported better at frequencies less than 0.1 Hz, at which efficiencies of ~100% are achieved (see Figure 6c). At higher frequencies, the particles move faster, and based on Stock’s law ($\vec{F}=6\pi \eta_{f}r_{p}$, where *F*, *η_f_*, and *r_p_* are the drag force, media viscosity, and particle radius, respectively), the drag force applied from the surrounding media to the particle increases. Hence, at higher frequencies, the particles no longer can follow the energy wells and have difficulties moving along the magnetic tracks.

By increasing the magnetic field strength, deeper energy wells are created (see Figure 6d). Hence, the resulting magnetic energy can overcome stronger drag forces, which means the system can operate at higher frequencies. In Figure 6e, the average particle transport speeds at different frequencies for various field strengths are plotted. As expected, the results indicate that faster particle transports at higher operating speeds can be achieved at stronger magnetic fields. At frequencies higher than a critical frequency, the particles no longer follow the external magnetic field, and the velocity drops. At frequencies below the critical frequency, a linear relationship between the particle velocity (or the particle current) and the applied frequency is seen. This linear relationship ($I_{P}=f/R_{m}$, where *I_P_*, *f*, and *R_m_* stand for the particle current, driving frequency, and magnetic resistivity, respectively) behaves analogously to Ohm’s law (I = V/R_e_, where I, V, and R_e_ stand for the electrical current, voltage, and electrical resistivity, respectively) in electrical circuits. Based on these experimental results, the optimum magnetic field cone angle, frequency, and strength suggested ranges are α = 45–50°, 0.05–0.1 Hz, and ~80–100 Oe, respectively.

We also studied the effect of the ratio of the particle size to the magnetic track gap size by defining the dimensionless parameters $\beta_{x}=r_{P}/x_{G}$ and $\beta_{y}=r_{P}/y_{G}$, where *x_G_*_,_ and *y_G_* stand for the gap size in the x and y directions, respectively (see Figure 1). Appropriate open trajectories were observed for dimensionless ratios, $\beta_{x}\geq 0.32$ and $\beta_{y}\geq -0.84$ (A $\beta_{y}$ with a minus sign stands for the case where the I bar is shifted in the -y direction). The simulation results in Figure 7 for small $\beta_{x}$ (i.e., $\beta_{x}<0.32$) demonstrate an energy barrier between the C-shaped magnet and the next I bar, which prevents particle movement between them. In Figure 7a, the particle is at point B. By rotating the magnetic field, as shown in Figure 7b, the particle is still at point B, between the two energy barriers. This situation is valid until the energy barrier located before the I bar disappears. At this field angle, as seen in Figure 7c, the magnetic particle sees an energy barrier in the forward path and a negative energy barrier in the backward path, and thus it moves back toward the C-shaped magnet on the left. Moreover, in the case of $\beta_{y}<-0.84$ in which the I bar is shifted downward, the particle on the I bar tip, as opposed to moving towards the next C-shaped magnet, moves along the I bar (see Figure 8, where the particle is initially located at point B on the I bar tip, and by rotating the magnetic field, ultimately it moves along the I bar in the +Y direction in Figure 8d).

Thus, shifting the I bars in the −y direction can be problematic. The I bars are considered in the design to hold the particle until the next energy well appears on the next C-shaped magnet to capture it and move it forward. When the I bars are shifted in the +y direction, at some point the energy well on their tips is too far from the particle on the C-shaped magnet (results not shown here). This particle will see a closer energy well on the next C-shaped magnet. Thus, if the particle can be held on the first C-shaped magnet for a while and if the distance between the two adjacent C-shaped magnets is short enough, the particle can be transported between them directly. Hence, we came up with the idea of removing the I bars and designing a pure C-shaped magnetic track. We performed the required simulations and good results were achieved (see Figure 9).

Figure 10 illustrates examples of experimental particle trajectories captured under a microscope. In Figure 10a, a sample CI design is shown. Three different particles are transported simultaneously. In this design, two different gap sizes are employed. We show that a design with gaps of zero (i.e., connected magnetic patterns) that agrees with our proposed design criteria (i.e., $\beta_{x}\geq 0.32$) also works fine. This achievement is promising since the fabrication of small gaps may be challenging with typical optical lithography/metal lift-off processes. Although we did not study it systematically, we keep the width of the magnetic patterns within the small range possible to be created with the conventional fabrication methods. That is because lowering the aspect ratio (i.e., length-to-width ratio) will make the behavior of this design more similar to the magnetic disks in the previous works (see Figure 2), which have problems in manipulating particles in tri-axial magnetic fields. Moreover, we fabricated thin films with thicknesses of 100 and 200 nm, both of which showed similar results. Since further increasing the film thickness may result in the forming of physical barriers that may alter the particle trajectories, we keep the film thickness in this range.

In Figure 10b, the I bars are removed, and the device works well, as predicted. Appendix A shows the particle movement in this pattern too. In Figure 10a,b, magnetic particles are used to perform the experiments. To demonstrate the application of the introduced C-shaped magnetic track geometries, we study the behavior of magnetically labeled AML cells on this magnetic pattern. The blue dotted lines in this figure represent sample particle and cell trajectories. In the experiments, we did not observe any particle pairs, which shows using a tri-axial field in the proposed design is promising for single-particle studies.

As a pilot test, we used the proposed chip to move oligo (dT)-coupled beads for capturing mRNA in an mRNA-containing solution. This protocol is based on pairing the poly-A tail of mRNA with the oligo dT sequences available on the beads. The beads need time to form the pairs and collect the mRNAs. Hence, after introducing the beads to the chip, we applied external rotating magnetic fields at various speeds to move the particles. To give the beads enough time, we ran the experiments at 0.01, 0.02, and 0.05 Hz. When the beads reach the region with mRNAs (see the rectangular area in the schematic illustrated in Figure 10d), they start collecting the mRNAs. At lower speeds, they have more time to pair with the mRNAs, and they are supposed to collect more mRNAs. The resulting normalized captured mRNA levels for the three different driving frequencies are plotted in Figure 10e. These results confirm higher mRNA capture levels at lower speeds, as expected. In conventional mRNA collection based on magnetic beads, an external magnet is used to collect the magnetic beads from the suspension. However, in the method proposed here, the beads move out of the sample suspension by the magnetophoretic circuits. This pilot experiment shows the proposed device has good potential in bio applications.

## 4. Conclusions

In this work, we proposed and studied a C-shaped magnetic track to transport single magnetic particles and magnetized cells in a microfluidic environment. This simple design transports particles in a tri-axial magnetic field, in which a repulsive force is applied between the particles to prevent particle cluster formation. This behavior is similar to the electron-electron repulsion seen in electronic circuits with no undesired charge packet build-up. Additionally, the vertical magnetic field bias prevents the degeneracy in the clock cycle seen in magnetophoretic circuits operating in two-dimensional in-plane magnetic fields and offers unique positions for the individual particles.

We performed a complete study on various parameters, including the magnetic field intensity, magnetic field cone angle, driving frequency, particle size, and design geometries. We showed that magnetic field cone angles in the range of α = 45–50°, driving frequencies in the range of 0.05–0.1 Hz, and magnetic field intensities of ~80–100 Oe are good choices to achieve proper device operation. We demonstrated that at higher frequencies or weaker magnetic fields, the particles no longer follow the external field, and the device performance drops. We also showed that at low frequencies, the particle velocity is proportional to the driving frequency, a behavior similar to Ohm’s law in electrical systems, where the electron current is proportional to the driving voltage. We also found that for proper particle transport, $\beta_{x}\geq 0.32$ and $\beta_{y}\geq -0.84$ are needed, which means particles have difficulties moving over large magnetic track gaps.

We showed that although the I bars in the CI pattern hold the particles and transfer them to the next C-shaped magnet, pure C-shaped designs work fine too. We also showed in a pilot biological study that barcode-carrying beads can capture mRNAs at rates proportional to their transport speed on the chip. Flawless parallel transport of magnetic beads and magnetized living cells using the proposed magnetic pattern was demonstrated. In microfluidic chips based on the proposed circuits, a large number (e.g., thousands) of single cells and particles can be manipulated simultaneously without any additional complexity. The chip can be used to form arrays of single cells to study their dynamic behavior (e.g., cytokine secretion profiles, drug responses, etc.). The proposed design offers smooth particle transport on simple magnetic paths and prevents undesired particle cluster formation, which opens the windows to important applications in the fields of lab-on-a-chip and single-cell biology.

## Figures and Tables

**Figure 1 micromachines-13-02177-f001:**
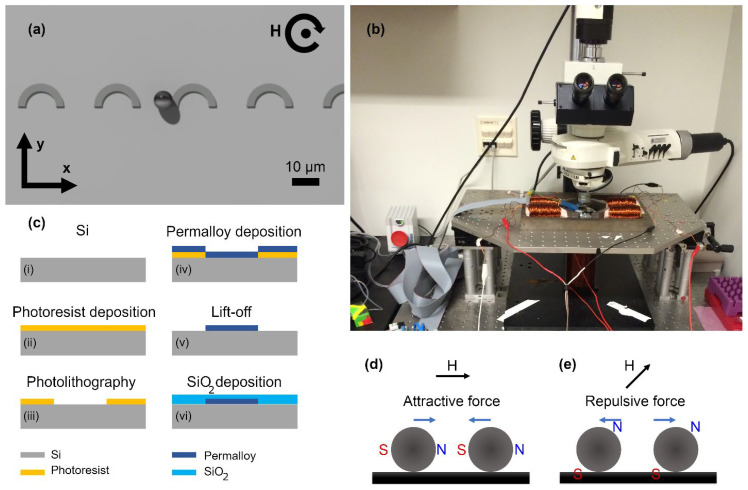
(**a**) A schematic of the proposed C-shaped design is illustrated. The little sphere represents a particle. The circular arrow and the dot in the middle stand for the in-plane rotating field and the vertical bias field, respectively. (**b**) The experimental setup is shown. (**c**) A schematic of the fabrication steps is presented. (i) We start with a silicon wafer. (ii) A photoresist layer is deposited on wafers. (iii) Using photolithography, the photoresist layer is patterned. (iv) Then a thin layer of permalloy evaporated onto the surface. (v) Next, after a liftoff process the unwanted metal is removed. (vi) Finally, the chip is coated with a thin layer of SiO_2_. (**d**) In an in-plane field, the force between magnetic particles is attractive (side view). (**e**) Adding a vertical bias field to the in-plane field, turns the force between magnetic particles into a repulsive force (side view). In (**d**,**e**), the circles and the black rectangle stand for the particles and the substrate, respectively. The black and blue arrows stand for the external magnetic field and the force directions, respectively. N and S represent the north and south poles of the magnetic dipole moment.

**Figure 2 micromachines-13-02177-f002:**
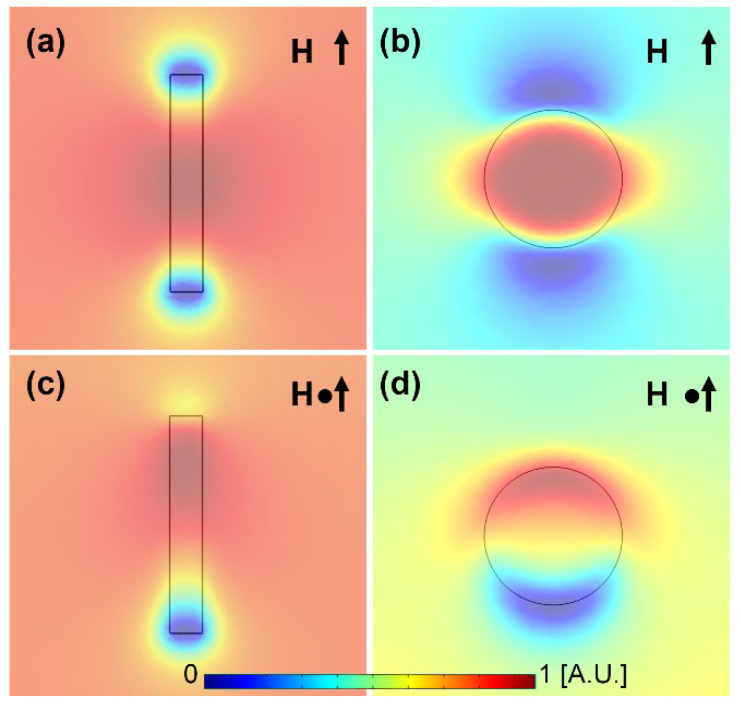
Magnetic energy simulation results for (**a**) a magnetic bar and (**b**) a magnetic disk exposed to an external in-plane magnetic field, and (**c**,**d**) after superimposing a vertical bias magnetic field perpendicular to the plane of the magnetic thin films are shown. The blue and red areas stand for the regions with low and high magnetic energies. The black arrows depict the magnetic field direction, and the little black dots represent the vertical bias field.

**Figure 3 micromachines-13-02177-f003:**
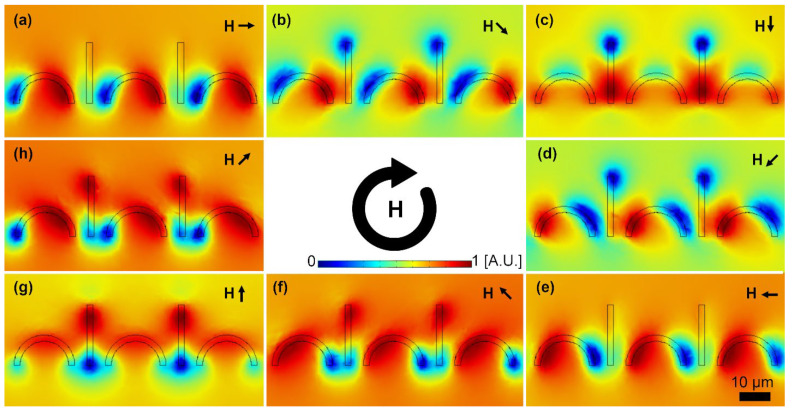
The potential energy distributions for various magnetic field angles, when the field rotates a cycle, are presented. In this figure, the small black arrows stand for the magnetic field directions. The field is rotating clockwise and the steps (**a**–**h**) are 45°. The energies are plotted at the center of a particle with a radius of 2.5 μm. The blue and red areas indicate the regions with low and high potential energies, respectively.

**Figure 4 micromachines-13-02177-f004:**
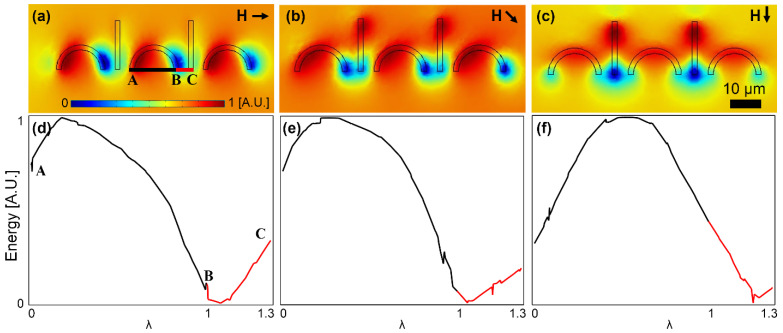
The particle transport from a C-shaped magnet to the next I bar is studied. (**a**–**c**) The potential energy distributions for three sequential magnetic field directions are illustrated. (**d**–**f**) The energies along the lines AB and BC (see (**a**)) for the field directions of (**a**–**c**) are plotted in (**d**–**f**), respectively. The small black arrows stand for the magnetic field directions. The field is rotating clockwise, and the steps (**a**–**c**) are 45°. The energies are plotted at the center of a particle with a radius of 2.5 μm. The blue and red areas indicate the regions with low and high potential energies, respectively.

**Figure 5 micromachines-13-02177-f005:**
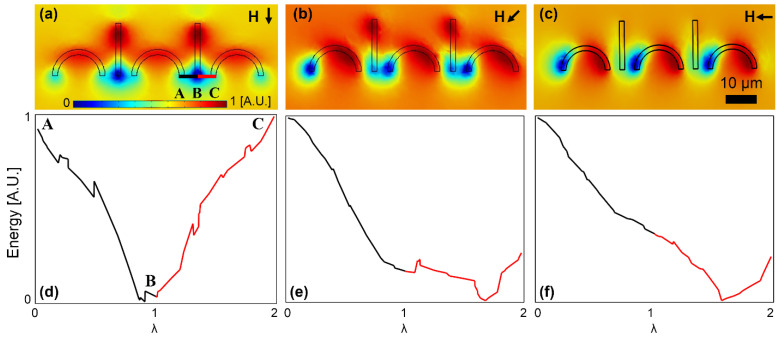
The particle transport from the I bar tip to the next C-shaped magnet is studied. (**a**–**c**) The potential energy distributions for three sequential magnetic field directions are illustrated. (**d**–**f**) The energies along the lines AB and BC (see (**a**)) for the field directions of (**a**–**c**) are plotted in (**d**–**f**), respectively. The small black arrows stand for the magnetic field directions. The field is rotating clockwise, and the steps (**a**–**c**) are 45°. The energies are plotted at the center of a particle with a radius of 2.5 μm. The blue and red areas indicate the regions with low and high potential energies, respectively.

**Figure 6 micromachines-13-02177-f006:**
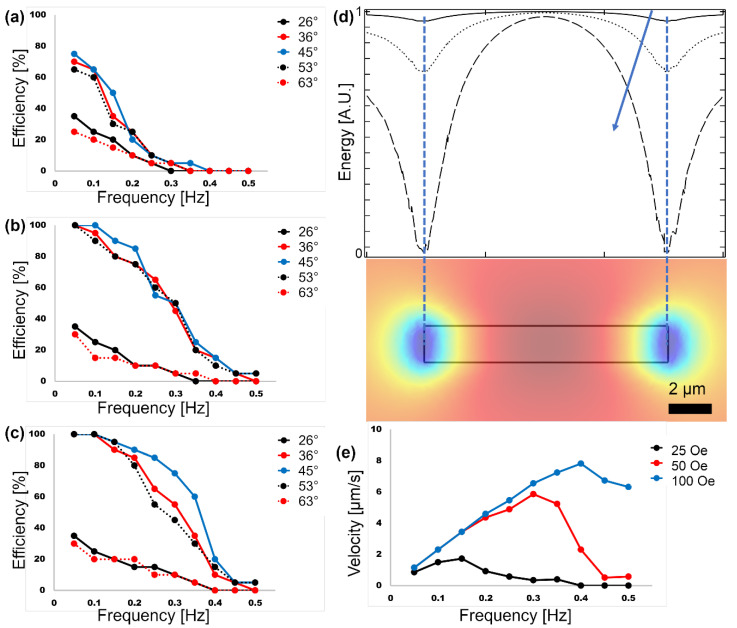
The particle transport efficiencies (i.e., the number of periods at which the particles move flawlessly) as a function of frequency for various cone angles are plotted. The experiments are repeated for magnetic field strengths of (**a**) 25 Oe, (**b**) 50 Oe, and (**c**) 100 Oe. In these experiments, the particle mean diameter is 8.4 µm. The field cones are α = 26° (solid black), α = 37° (solid red), α = 45° (solid blue), α = 53° (dotted black), and α = 63° (dotted red). (**d**) The energy along the I bar main axis when exposed to the magnetic field along that axis for magnetic field strengths of 25 Oe (solid line), 50 Oe (dotted line), and 100 Oe (dashed line) are plotted. The blue arrow indicates increasing magnetic field strength. (**e**) The particle velocities versus frequency at magnetic fields with various strengths are plotted.

**Figure 7 micromachines-13-02177-f007:**
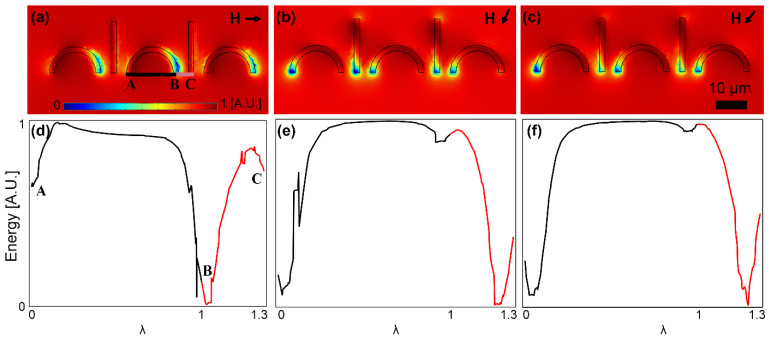
The particle transport failure from the I bar tip, in an unsuitable *β_x_* = 0.2, is studied. (**a**–**c**) The potential energy distributions for three sequential magnetic field directions are illustrated. (**d**–**f**) The energies along the lines AB and BC (see (**a**)) for the field directions of (**a**–**c**) are plotted in (**d**–**f**), respectively. The small black arrows stand for the magnetic field directions. The field is rotating clockwise. The energies are plotted at the center of a particle with a radius of 0.5 μm. The blue and red areas indicate the regions with low and high potential energies, respectively.

**Figure 8 micromachines-13-02177-f008:**
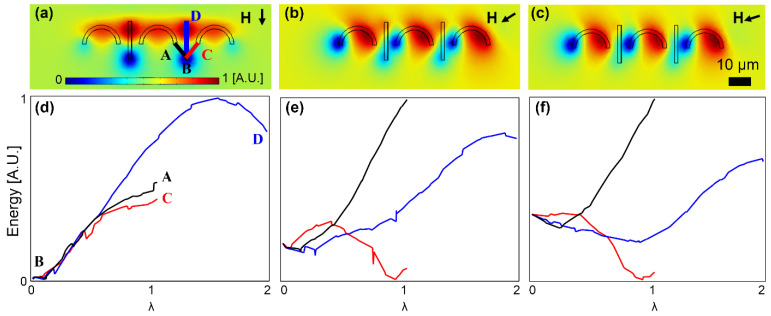
The particle transport failure from the I bar tip, in an unsuitable *β_y_* = −1.52, is studied. (**a**–**c**) The potential energy distributions for three sequential magnetic field directions are illustrated. (**d**–**f**) The energies along the lines AB and BC (see (**a**)) for the field directions of (**a**–**c**) are plotted in (**d**–**f**), respectively. The small black arrows stand for the magnetic field directions. The field is rotating clockwise. The energies are plotted at the center of a particle with a radius of 2.5 μm. The blue and red areas indicate the regions with low and high potential energies, respectively.

**Figure 9 micromachines-13-02177-f009:**
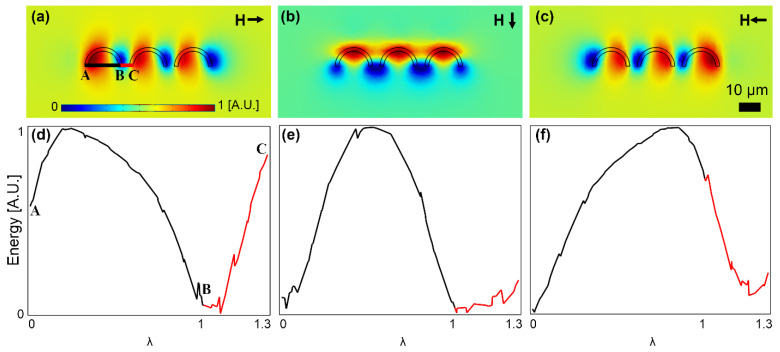
The particle transport on a pure C-shaped magnetic track is studied. (**a**–**c**) The potential energy distributions for three sequential magnetic field directions are illustrated. (**d**–**f**) The energies along the lines AB and BC (see (**a**)) for the field directions of (**a**–**c**) are plotted in (**d**–**f**), respectively. The small black arrows stand for the magnetic field directions. The field is rotating clockwise, and the steps (**a**–**c**) are 90°. The energies are plotted at the center of a particle with a radius of 2.5 μm. The blue and red areas indicate the regions with low and high potential energies, respectively.

**Figure 10 micromachines-13-02177-f010:**
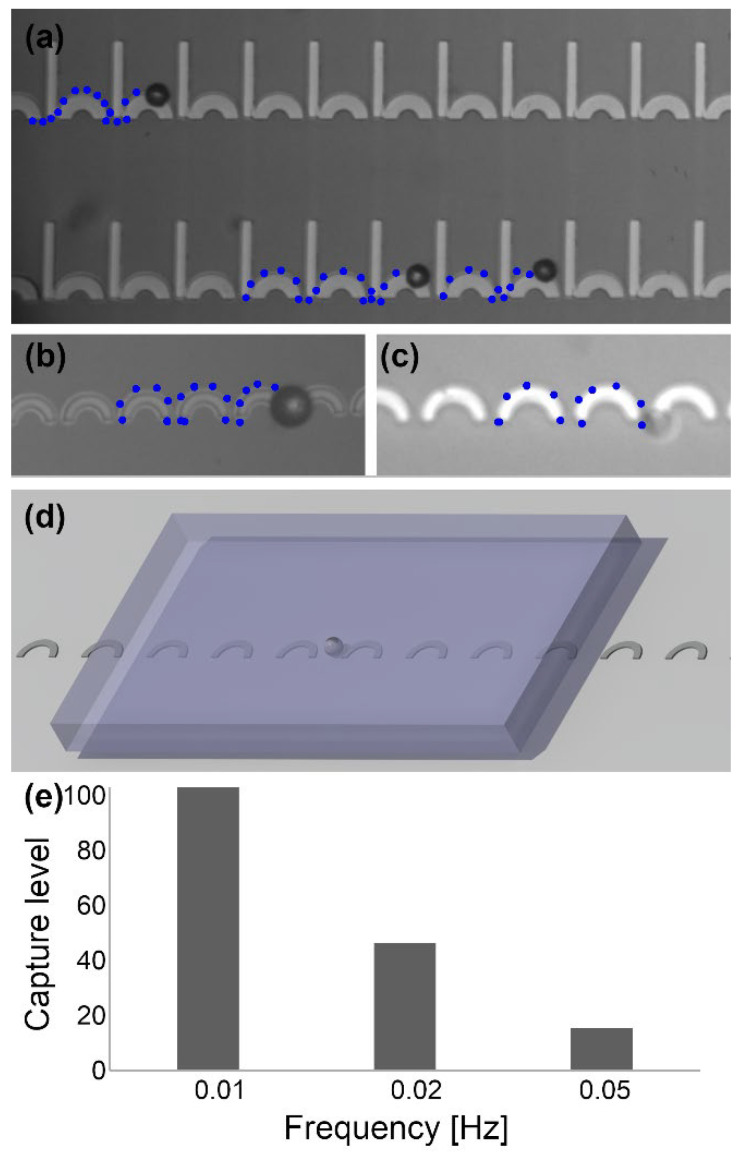
Sample microscopy images of particles transported along the (**a**) CI magnetic track and (**b**) C-shaped magnetic track are illustrated. (**c**) Magnetically labeled AML cells are transported on the proposed C-shaped magnetic track. A magnetic field with a conical angle of α = 45° (i.e., the in-plane and vertical field components are fixed at 50 Oe) at a frequency of 0.05 Hz, which is rotating clockwise is applied to the chip. The blue dotted lines depict the trajectories of the particles and cells. (**d**) A schematic showing a particle moving on the proposed magnetic track in an area containing the solution of interest. (**e**) mRNA capture levels for barcode-carrying magnetic beads being transported at various deriving frequencies.

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
