# Peer review of "Controlled Transport of Magnetic Particles and Cells Using C-Shaped Magnetic Thin Films in Microfluidic Chips"

_micromachines, 2022, doi:10.3390/mi13122177_

Round 1

Reviewer 1 Report

In this study, the author proposed a C-shaped magnetic thin film to control the transport of the magnetic particles and cells. This system enables magnetic particles to repel each other to prevent the formation of unwanted clusters and to control particle transport. The article is very innovative but some mistakes should be modified:

1. Context logic should be strengthened in this article.

2. The pictures provided are not detailed enough.

3. The experiment is not shown clearly in this paper.

Reviewer 2 Report

The authors present numerical simulations and experimental characterization of a microfluidic platform with C-shaped magnetic thin films operating in a tri-axial rotating magnetic field to precisely transport magnetic particles and cells, with potential for single cell analysis applications. Although the presented work has potential, the novelty when compared to other magnetic systems needs to be better explained, and the authors should be more careful in the writing and formatting (standard sentences from the template, missing references, etc.). Additionally, there is insufficient experimental data, and the experimental methods are unclear.

My comments, questions and concerns are listed below:

Introduction section: there are already reported in literature many works using magnetic fields for particle and cells transport. How does this work differ from the works of other authors? The novelty of the work needs to be better explained.

Page 2, fig 1. a) include the dimensions of each C-shaped magnet.

Page 3, line 83: in the start of section 2, it appears: “The Materials and Methods should be described with sufficient details to allow”. Please remove this sentence.

Page 3, section 2. Please add references in the theory (equations 1 and 2).

The description of the numerical model needs more detail, so the model can be replicated by other researchers/readers and the readers can fully understand the results presented in the results section (figures 2, 3, etc). Regarding the COMSOL Model, the authors need to specify:

- the governing equations/models implemented;

- boundary and initial conditions;

- solvers (are they stationary or time dependent, direct or iterative?)

- Is the model implemented in a 2D or 3D domain (it is not clear)? Can a 2D simulation fully represent the 3D behavior?

- materials: was the silicon substrate also considered in the simulations?

- geometries and dimensions.

Regarding the mesh, the authors refer a mesh with 212364 elements. But I assume that is only for the C-shaped magnets, right? It should be referred, as probably the bar and disk geometries presented in Figure 2 were simulated based on a different number of mesh elements.

Page 4, line 129: what do the authors mean with the sentence “We assume that permalloy (i.e., Ni80Fe20) is a soft magnetic material deposited with the desired shape on the proposed chips”?  The results are based on assumptions or experimental verification? Didn’t the authors characterize/verify the shape and magnetic properties of the deposited films?

Page 4, line 132 – “Add reference from TI Paper”. It seems that a reference is missing.

Results section: Figures 2, 3, 4, 5, etc (all the figures with simulation plots): the authors refer that the color in the simulation results represent the energy (low – blue; high – red). However, the color bar ranges from 0-1, with non-dimensional units. Why presenting as a non-dimensional variable instead of the absolute energy?

In the plots of figures 4, 5, 7, 8, 9, please add scale bars so the readers can understand the distances between each point (A, B, C or D, in the case of fig 8). Also, the analyzed span axis should be detailed with the distance values, and the same for energy in y axis.

In page 8, fig 6 d), what does the blue arrow represent?

Figure 6 d), energy plot should present values in y axis.

How did the authors define/select the dimensionless parameter beta_x and beta_y? Did the authors perform dimensional analysis of the problem or defined those parameters empirically?

Did the authors study the effect of the thickness/width of the C-magnets?

The experimental results are very incomplete, as figure 10 a) and b) only presents a few beads and does not allow to assess the particle tracking. I suggest the authors to add a video of the beads’ movement as supplementary material, to help understand the particles movement along the track. Additionally, authors should present some result quantification, for instance, including plots of the beads position in x over time (or other quantification methods), instead of random microscope figures.

The authors also include a plot showing a mRNA-capturing experiment with barcode-carrying magnetic beads. However, regarding this study, only a plot is presented, with no further results nor discussion. How did the authors perform this assay? The methods should be clear. How was capture level defined? this part is really incomplete and needs to be considerably improved, as the results aren’t enough to demonstrate the claimed conclusions.

Reviewer 3 Report

Simulations with multiple magnetic beads are missing; why should they repel each other? This is not clear to me. Did you observe this?
Furthermore, an experimental part is missing before the results and discussion section.
I addition, it is not clear to me why the C-shape was chosen and the advantage regarding previous arragements.
Please describe how the cells were magnetically labelled and why data on teh dynamic behavior of different cells are important.

Round 2

Reviewer 1 Report

A C-shaped magnetic thin film to control the transport of the magnetic particles and cells has been proposed. The author modified the pictures and enriched the experimental content, and the article is of great research value.

Reviewer 2 Report

The authors addressed all my comments satisfactorily. The manuscript was significantly improved.

Reviewer 3 Report

thank you for the fast und extensive rework of the paper und much success for you ongoing research!